# High power rechargeable magnesium/iodine battery chemistry

Huajun Tian[1,2,*], Tao Gao[1,*], Xiaogang Li[1], Xiwen Wang[1], Chao Luo[1], Xiulin Fan[1], Chongyin Yang[1], Liumin Suo[1], Zhaohui Ma[1], Weiqiang Han[2,3] & Chunsheng Wang[1]

Rechargeable magnesium batteries have attracted considerable attention because of their potential high energy density and low cost. However, their development has been severely hindered because of the lack of appropriate cathode materials. Here we report a rechargeable magnesium/iodine battery, in which the soluble iodine reacts with $Mg^{2+}$ to form a soluble intermediate and then an insoluble final product magnesium iodide. The liquid–solid two-phase reaction pathway circumvents solid-state $Mg^{2+}$ diffusion and ensures a large interfacial reaction area, leading to fast reaction kinetics and high reaction reversibility. As a result, the rechargeable magnesium/iodine battery shows a better rate capability (180 mAh g$^{-1}$ at 0.5 C and 140 mAh g$^{-1}$ at 1 C) and a higher energy density ($\sim$400 Wh kg$^{-1}$) than all other reported rechargeable magnesium batteries using intercalation cathodes. This study demonstrates that the liquid–solid two-phase reaction mechanism is promising in addressing the kinetic limitation of rechargeable magnesium batteries.

[1] Department of Chemical and Biomolecular Engineering, University of Maryland, College Park, College Park, Maryland 20740, USA. [2] Ningbo Institute of Materials Technology and Engineering, Chinese Academy of Sciences, Ningbo 315201, China. [3] School of Materials Science and Engineering, Zhejiang University, Hangzhou 310027, China. * These authors contributed equally to this work. Correspondence and requests for materials should be addressed to H.T. (email: tianhuajun@nimte.ac.cn) or to W.H. (email: hanwq@zju.edu.cn) or to C.W. (email: cswang@umd.edu).

High abundance (1.5 wt% in earth crust and 0.13 wt% in sea water), high volumetric energy density (3,833 mAh cm$^{-3}$), very negative reduction potential ($-2.37$ V versus Standard Hydrogen Electrode) and, most importantly, highly reversible dendrite-free deposition, have made magnesium (Mg) metal an ideal anode material for post lithium ion battery chemistries[1,2]. The key to a high-performance rechargeable magnesium battery (RMB) lies in high voltage electrolytes and high voltage/capacity cathodes. In the past decades, although significant advances have been made in electrolyte development[3–5], little progress has been achieved in the cathode material study. The hurdle is the clumsy Mg$^{2+}$ intercalation because of the sluggish solid-state diffusion of the divalent Mg$^{2+}$ and the slow interfacial charge transfer. Compared with monovalent cations (Li$^+$, Na$^+$, K$^+$ and so on), the high charge density of Mg$^{2+}$ (twice as high as Li$^+$) inevitably raises the energy barriers for breaking its solvation sheath or ion-ligand pair upon interfacial charge transfer[6,7]. It also induces strong Coulombic interactions with the host upon ion insertion and hopping[8] and causes difficulty for the host to accommodate electrons[8,9]. As a result, most reported cathode materials show inferior kinetics and poor reversibility, except for Chevrel phase (Mo$_6$S$_8$)[10]. However, Mo$_6$S$_8$ only provides a limited energy density ($<126$ Wh kg$^{-1}$) because of its low operation voltage ($\sim1.2$ V) and low specific capacity (110 mAh g$^{-1}$)[10].

Extensive efforts have been conducted to improve the reaction kinetics and cycling stability of RMBs. The most successful strategy is to couple Li$^+$ intercalation cathodes (for example, TiS$_2$, LiFePO$_4$ and so on) with a Mg anode in a hybrid Mg$^{2+}$/Li$^+$ electrolyte so that the clumsy Mg$^{2+}$ intercalation can be partially or even completely replaced by fast Li$^+$ intercalation[11–14]. However, the energy density of such a hybrid battery is restricted by the Li salt concentration of the electrolyte that limited its practical application. There are also studies reporting that water can stimulate Mg$^{2+}$ intercalation and thus dramatically enhance the intercalation kinetics by shielding the charge of Mg$^{2+}$ via a solvent co-intercalation mechanism[15–17], yet the compatibility of Mg anode with water remains a problem. Up until now, there has been no facile strategy that could effectively tackle the Mg$^{2+}$ intercalation issue. A conversion cathode that undergoes two-phase (solid–liquid or liquid–solid) reaction pathway during an electrochemical reaction that does not rely on solid-state Mg$^{2+}$ diffusion is considered a promising direction[9]. The most successful demonstration of this

concept is Li/S batteries where a solid–liquid–solid reaction pathway takes place during the reduction of sulfur[18]. For Mg batteries, iodine (I$_2$) serves as a perfect cathode material for illustrating this concept. This is because I$_2$ and its partial reduction product, Mg(I$_3$)$_2$, have high solubility in ether-based electrolytes, but its final reduced product, MgI$_2$, is insoluble (Table 1). Thus, an I$_2$ cathode is expected to have faster redox reaction kinetics than intercalation cathodes. In addition, I$_2$ is able to provide a much higher voltage (2.1 V) and capacity (211 mAh g$^{-1}$) than Mo$_6$S$_8$ (Fig. 1a and Supplementary Table 1, see Calculation section).

Herein, we demonstrate a rechargeable Mg/I$_2$ battery that is able to provide capacity close to the theoretical value ($\sim200$ mAh g$^{-1}$) with an average voltage of 2.0 V at C/4, corresponding to an energy density of 400 Wh kg$^{-1}$. Prolonged cycling shows an excellent stability at C/2 with capacity retention of 94.6% for 120 cycles. A liquid–solid two-phase reaction mechanism (Fig. 1b) was validated using spectroscopic and microscopic characterizations. Because of the fast Mg$^{2+}$ diffusion in the electrolyte, the ready electron access of iodine species and large interfacial reaction area, the Mg/I$_2$ battery shows superior rate capability (180 mAh g$^{-1}$ at 0.5 C and 140 mAh g$^{-1}$ at 1 C). This proof-of-concept Mg/I$_2$ battery demonstrates the feasibility of using a liquid–solid two-phase reaction route to address the challenging intercalation kinetics of RMB cathodes. The chemical insights obtained here can be beneficial for realizing an optimized system, a Mg/I$_2$ flow battery or other kinds of two-phase reaction RMB cathodes.

## Results

**Material preparation.** Because of the electronic insulating nature of I$_2$, it was intentionally impregnated into the pores of active carbon cloth (ACC) through a melt-diffusion method to enhance its electron access following previous reported method[19,20].

**Table 1 | Solubility of different iodine species in tetraglyme.**

| Solubility | I$_2$ | Mg(I$_3$)$_2$ | MgI$_2$ |
|---|---|---|---|
| Mass of solute per 100 ml solvent (g) | >75 | >100 | <0.1 |
| Molarity of atomic iodine (mol l$^{-1}$) | >5.9 | >7.6 | <0.007 |

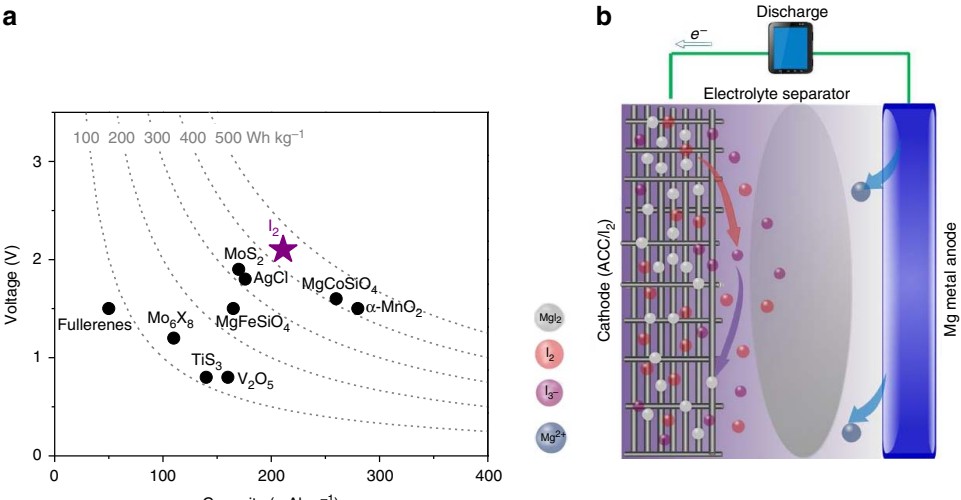

**Figure 1 | Schematic illustration of the rechargeable magnesium/iodine battery.** (**a**) The capacity and voltage of the iodine cathode compared with reported rechargeable magnesium batteries cathodes. (**b**) Schematic of rechargeable Mg/I$_2$ batteries.

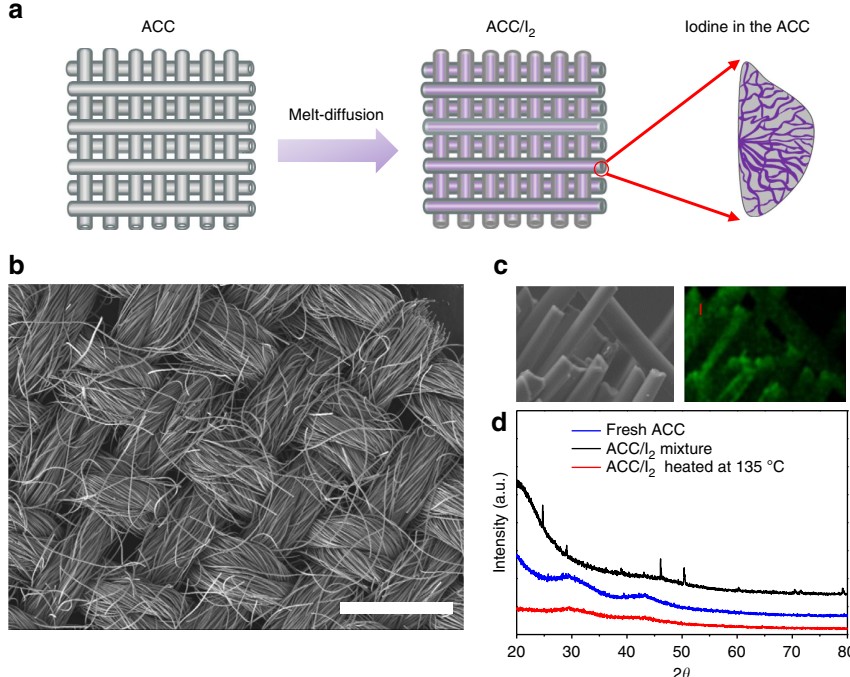

**Figure 2 | Material characterization of the active carbon cloth/iodine cathode.** (**a**) Procedure for preparing the ACC/$I_2$ electrode. (**b**) Scanning electron microscopy images of the ACC/$I_2$ cathode, scale bar: 1 mm. (**c**) Energy-dispersive spectroscopy mapping of I in the ACC/$I_2$ cathode. (**d**) X-ray diffraction pattern of the ACC/$I_2$ cathode.

Meanwhile, ACC can also inhibit the migration of dissolved $I_2$ towards the Mg anode because of the strong adsorption of $I_2$ in the ACC pores (Supplementary Fig. 1), thus mitigating the expected shuttle effect. The synthesis of the ACC/$I_2$ cathode is schematically shown in Fig. 2a. In brief, a mixture of ACC and $I_2$ was first sealed into an argon-filled container, and then heated to 135 °C for 12 h. At this temperature, the fluid $I_2$ will be infiltrated into the pores of the ACC through the capillarity effect (the melting point of $I_2$ is 113.7 °C). The weight ratio of $I_2$ in the composite electrode can be evaluated with thermogravimetric analysis. For an electrode with $I_2$ loading 2.8 mg cm$^{-2}$, the mass ratio of $I_2$ is ~27% (Supplementary Fig. 2). Scanning electron microscopy (SEM) images of the ACC/$I_2$ cathode demonstrates no residual $I_2$ on the surface (Fig. 2b,c), and a very uniform distribution of $I_2$ in carbon can be observed in the energy-dispersive spectrum of I (Fig. 2c). X-ray diffraction shows that the $I_2$ peak disappears after impregnation (Fig. 2d), suggesting that $I_2$ loses its long range order structure and is uniformly dispersed inside the pores of the carbon fiber.

**Electrochemical performance.** The electrolyte was synthesized by reacting magnesium bis(trimethylsilyl)amide ((HMDS)$_2$Mg) with aluminum chloride (AlCl$_3$) and magnesium chloride (MgCl$_2$) in tetraglyme (TEGDME) *in situ*. The preparation procedure is given in the Methods section. For convenience, the electrolyte is abbreviated as Mg-HMDS. The deposition/striping process of the electrolyte was measured by three-electrode cell and coin cell (Supplementary Figs 3 and 4) and a Coulombic efficiency of close to 100% can be obtained, consistent with previous work.[21] The electrochemical stability window of the electrolyte was also measured (Supplementary Fig. 3) and the electrolyte starts to show observable decomposition when voltage exceeds 2.7 V but vigorous oxidation did not happen until 3.0 V. The Mg/$I_2$ battery was assembled by coupling an ACC/$I_2$ disk, a glass fiber separator and an Mg foil anode into a Swagelok cell. The typical $I_2$ loading

of the ACC/$I_2$ disk in the electrochemical tests is ~1.0 mg cm$^{-2}$ and the electrolyte volume is 100 µl. A typical discharge/charge curve of the Mg/$I_2$ cell in 0.5 M Mg-HMDS electrolyte is shown in Supplementary Fig. 5. The open-circuit voltage of the Mg/$I_2$ cell is 2.25 V. During discharge, the ACC/$I_2$ cathode experiences a quick potential drop, and then reaches a plateau at ~1.95 V. The first discharge shows a capacity of ~310 mAh g$^{-1}$, corresponding to an $I_2$ utilization of ~94.7%, excluding the contribution from the ACC (the charge/discharge curve of a blank ACC is shown in Supplementary Fig. 6). The Coulombic efficiency is ~75%, suggesting the presence of shuttle effect during charging. As $I_2$ and Mg($I_3$)$_2$ are highly soluble in the electrolyte, it is inevitable to have $I_2$ loss during charge/discharge that causes the shuttle effect and results in low Coulombic efficiency. To mitigate the shuttle effect, an electrolyte with a high salt concentration (2 M Mg-HMDS) was used in the Mg/$I_2$ battery as concentrated electrolytes were proven effective for this purpose.[22] The discharge curves of the ACC/$I_2$ cathode in this concentrated electrolyte show a discharge behaviour similar to that in the low concentration electrolyte: a short plateau at 2.2 V followed by a long plateau at 1.9 V (Fig. 3a). However, a remarkable difference can be observed for the charge curves. In the high concentration electrolyte (2 M Mg-HMDS), the ACC/$I_2$ cathode shows a clear charge plateau at ~2.5 V with a potential spike at the end of the charge (Fig. 3a), whereas the ACC/$I_2$ cathode in the low concentration electrolyte (0.5 M Mg-HMDS) does not show any potential rising at the end of charge even when the capacity exceeded the theoretical value. The shuttle effect of polyiodide was thus effectively mitigated as evidenced from the significantly increased Coulombic efficiency (close to 100%). To understand the reaction mechanism of the Mg/$I_2$ chemistry, we performed cyclic voltammetry with the Mg/$I_2$ battery at a slow scan rate of 0.1 mV s$^{-1}$ (Fig. 3b). Two cathodic peaks at ~2.2 and ~1.8 V can be observed, corresponding to $I_2$/$I_3^-$ redox couple and $I_3^-$/$I^-$ redox couple, respectively. Two anodic peaks at 2.2 and 2.7 V can also be observed. Both the charge/discharge curve and

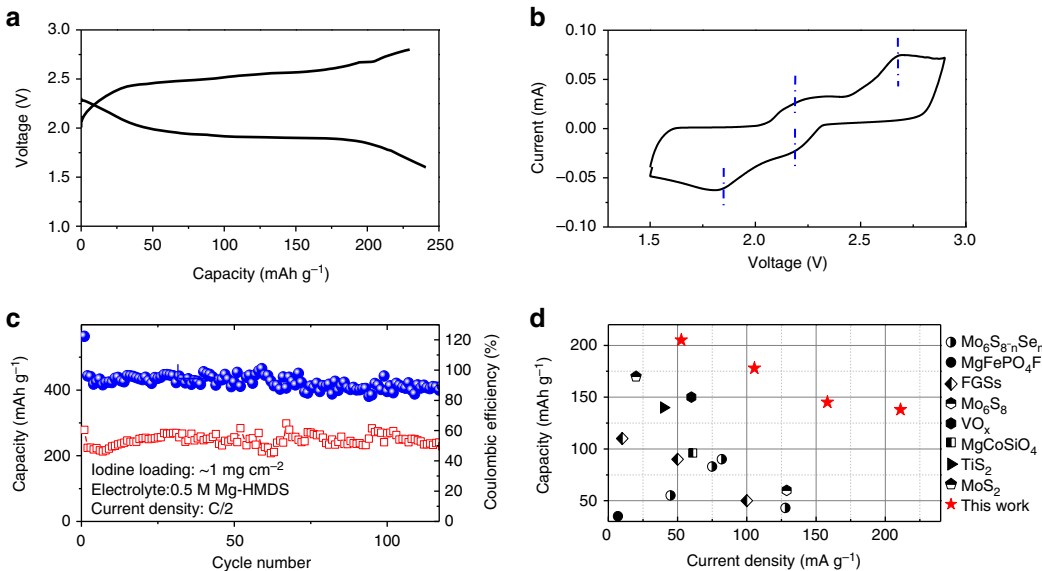

**Figure 3 | Electrochemical performance of the rechargeable Mg/I$_2$ battery.** (**a**) A typical discharge/charge curve of the Mg/I$_2$ battery with ACC/I$_2$ cathode. (**b**) Cyclic voltammogram of the Mg/I$_2$ battery with ACC/I$_2$ cathode. Scan rate: 0.1 mV s$^{-1}$. (**c**) Cycling stability of the Mg/I$_2$ battery at 0.5 C (105.5 mA g$^{-1}$) with ACC/I$_2$ cathode. (**d**) Rate capability of Mg/I$_2$ battery with ACC/I$_2$ cathode. The rate capabilities of other cathode materials are also plotted for comparison.

cyclic voltammogram prove the good reversibility of the I$_2$ redox couple in the Mg-HDMS electrolyte. As a result, the Mg/I$_2$ battery exhibits excellent long-cycle stability, with a high-capacity retention of 94.6% after 120 cycles at a rate of C/2, and could provide a specific capacity of ~180 mAh g$^{-1}$ at this rate (Fig. 3c). Most importantly, the Mg/I$_2$ battery exhibits a superior rate capability as illustrated in Fig. 3d, in which all RMB intercalation cathodes reported to date are plotted for comparison. It can even provide a specific capacity of 140 mAh g$^{-1}$ at high rate of 1C. Thus, the Mg/I$_2$ battery demonstrates significantly better rate performance than all RMB intercalation cathodes, especially at high current densities (>200 mA g$^{-1}$). This observation confirms our hypothesis that a two-phase conversion reaction can dramatically enhance the kinetics of RMBs.

**Two-phase reaction mechanism.** A series of microscopic and spectroscopic measurements were conducted to understand the reaction mechanism of the Mg/I$_2$ battery. We examined the solubility of different iodine species in ethereal solvents (Table 1). I$_2$ and Mg(I$_3$)$_2$ are highly soluble in TEGDME and the transparent solution quickly turned dark purple once a solute was added, whereas MgI$_2$ has negligible solubility in TEGDME and the MgI$_2$-TEGDME solution maintains transparent (Supplementary Fig. 7). A two-electrode flooded cell using ACC as a current collector, 0.15 M I$_2$ in Mg-HMDS electrolyte as catholyte and Mg foil as anode was assembled for *in situ* observation of the colour change of the catholyte during CCCV discharge (constant current and then constant voltage discharge) (Fig. 4a). As can be seen, the dark purple colour of the catholyte faded gradually during discharge as I$_2$ was continuously reduced, and the colour nearly disappeared when the cell was discharged to 1.3 V, indicating the soluble I$_2$/Mg(I$_3$)$_2$ species were almost entirely converted to the insoluble MgI$_2$. After a full discharge, the insoluble MgI$_2$ product could be observed in the cell. The colour change phenomena from I$_3^-$ to I$^-$ was also observed in related researches on I$_3^-$/I$^-$ redox[23]. Because of lack of standard Fourier transform infrared spectroscopy (FT-IR) peaks for I$_3^-$ species, the

FT-IR spectra of I$_2$, I$^-$ and I$_3^-$ were first characterized as references (Fig. 4b,c). The electrolyte was analysed at different states during discharge using calibrated FT-IR spectroscopy. During discharge, the peak at 1,044 cm$^{-1}$ was gradually enhanced because of the formation of I$_3^-$ (Fig. 4d,e), as the 1,044 cm$^{-1}$ peak had been observed in I$_3^-$ solution (Mg(I$_3$)$_2$ in TEGDME) because of C-O stretching vibrations[24]. In addition, a negative shift of two peaks of ~1,350 and 1,250 cm$^{-1}$ from 2.2 to 2.1 V was observed that is due to the formation of I$_3^-$ (ref. 25). Therefore, I$_2$ was reduced to I$_3^-$ after discharging to 2.1 V. After that, those two peaks began to positively shift upon further discharge to 1.5 V. The positive shift could be attributed to the reduction of I$_3^-$ to I$^-$ (Fig. 4b,c). The FT-IR results demonstrated that iodine undergoes reduction reaction from I$_2$ to I$_3^-$ and then to I$^-$ during discharge in the Mg/I$_2$ battery. Note that the morphology change of the carbon cloth in Fig. 4a is due to longtime immersing of carbon in the solution instead of iodine redox reaction, as ACC/I$_2$ cathode cycled in Swagelok cell showed no morphology change compared with fresh ACC (Supplementary Fig. 8).

X-ray photoelectron spectroscopy (XPS) was employed to examine the surface chemistry changes of the ACC/I$_2$ cathode and the Mg anode during discharge/charge at different states. The oxidation states of iodine in pristine ACC/I$_2$, fully discharged ACC/I$_2$ and fully charged ACC/I$_2$ were monitored through XPS (Fig. 5a). The high-resolution I 3d spectrum of the fresh ACC/I$_2$ cathode is mainly composed of elemental iodine, as evidenced by I 3d$_{5/2}$ peak located at 620 eV (ref. 26). Two extra I 3d$_{5/2}$ peaks located at 622.7 and 618.0 eV correspond to I-O bond and I-C bond, respectively[26] that may come from the electrode preparation process. After a full discharge to 1.3 V, the I 3d$_{5/2}$ peak shifts to 619.2 eV, indicating that I$_2$ has been reduced to a lower oxidation state (I$^-$). This result confirms the formation of MgI$_2$ as the fully discharged product. After a full charge to 2.8 V, I 3d$_{5/2}$ peak shifts back to 620.2 eV, suggesting oxidation of MgI$_2$ back to elemental iodine. High-resolution Mg 1s spectrum shows a peak shift from 1,303.2 to 1,303.8 eV after discharge, indicating an increase in Mg oxidation state (Fig. 5b). The I 3d spectrum of the Mg anode after discharge evidences the formation of a MgI$_2$

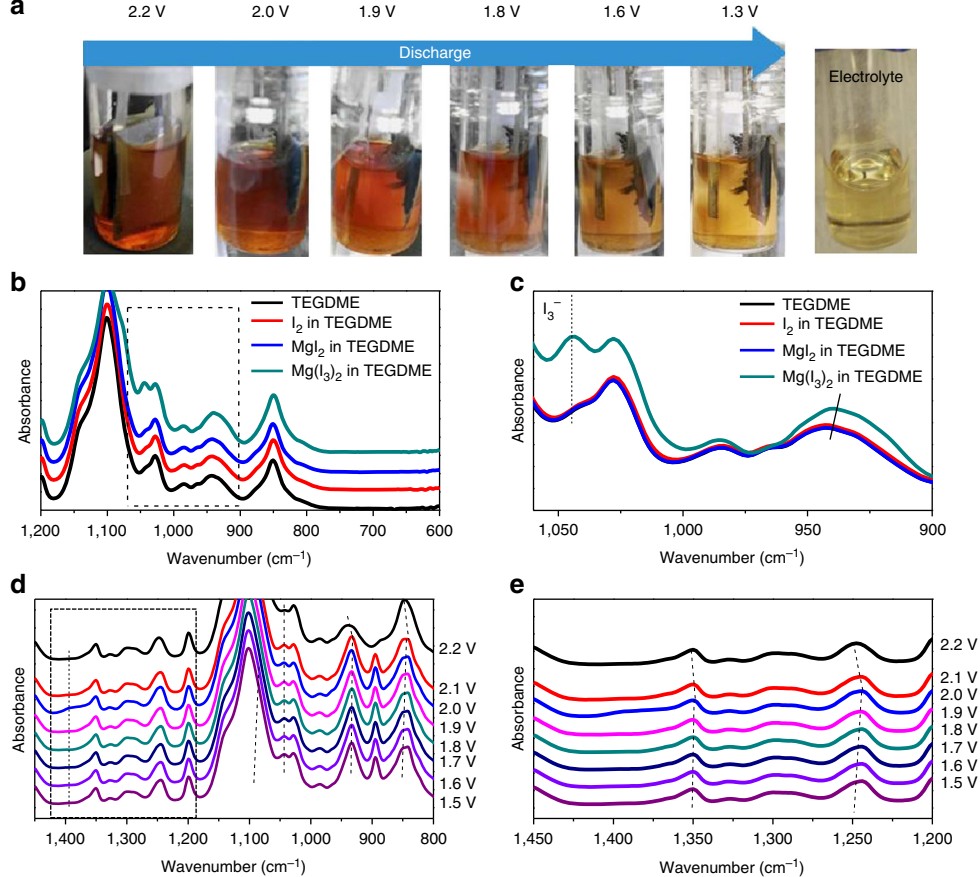

**Figure 4 | Fourier transform infrared (FT-IR) spectroscopy study of the electrolyte.** (**a**) Visual images of the discharge process of a Mg/$I_2$ battery at different discharge stages. (**b**) The controlled FT-IR spectra of $I_2$, $MgI_2$ and $Mg(I_3)_2$ in tetraglyme. (**c**) Magnified view of the regions outlined in **b**. (**d**) The *ex situ* FT-IR spectra of Mg/$I_2$ cell during the discharge process. (**e**) Magnified view of the regions outlined in **d**.

layer because of the expected shuttle effect (Fig. 5c). We measured the ionic/electronic conductivities of $MgI_2$, and the results show it has an ionic conductivity of $\sim 2.0 \times 10^{-5}\,\mathrm{S\,cm^{-1}}$ and an electronic conductivity of $\sim 2.1 \times 10^{-9}\,\mathrm{S\,cm^{-1}}$, consistent with previous report[27]. Therefore, the formed $MgI_2$ layer is likely to function as a solid electrolyte interface that can prevent further reaction of iodine species with Mg. A small elemental $I_2$ peak is also observed on the Mg anode after discharge, probably because of the disproportion of $Mg(I_3)_2$ in the residual electrolyte during sample preparation. Combining the XPS results with the FT-IR results, we can propose the following mechanism for the rechargeable Mg/$I_2$ batteries:

$$3I_2 + 2e^- + Mg^{2+} \rightarrow Mg(I_3)_2 \quad 2.8-2.0\,\mathrm{V} \quad (1)$$

$$Mg(I_3)_2 + 4e^- + 2Mg^{2+} \rightarrow 3MgI_2 \quad 2.0-1.3\,\mathrm{V} \quad (2)$$

Electrochemical impedance spectroscopy tests were conducted to examine the charge transfer kinetics of the Mg/$I_2$ battery (Supplementary Figs 9–11). As can be expected, the charge transfer resistance of the Mg/$I_2$ battery is one order lower than that of a rocking-chair battery ($Mo_6S_8$/Mg). This can be explained by the significantly increased quantity of reaction sites in the Mg/$I_2$ battery, as the liquid–solid two-phase reaction can theoretically take place anywhere on the carbon–electrolyte interface. In contrast, in a rocking-chair battery, reactions mainly occur on the three-phase interface (active material–carbon–electrolyte), where the active material can readily access both electrons and ions.

The above experimental results have also confirmed our hypothesis that $I_2$ undergoes a liquid–solid two-phase reaction in the Mg/$I_2$ battery during reduction. This two-phase reaction is highly reversible and offers remarkable reaction kinetics because of the bypass of solid-state $Mg^{2+}$ diffusion and the large interfacial area for charge transfer reaction. Because of the high solubility of $I_2$ and polyiodide, the inevitable shuttle effects leads to low Coulombic efficiency and passivation of the Mg anode. This property of the $I_2$ redox couple is very similar to that of sulfur in Li/S chemistry. As the shuttle effect in Li/S chemistry can be significantly mitigated by tailoring the physical and chemical properties of the sulfur host[19,28–33], we believe that the polyiodide shuttle effect can also be substantially prevented through host optimization. We show in our preliminary experiment that the shuttle effect can be greatly mitigated by using microporous carbon (MPC) with smaller pore size ($\sim 0.5\,\mathrm{nm}$) (morphology shown in Supplementary Fig. 12; discharge/charge of blank MPC shown in Supplementary Fig. 13)[34]. The results show clear discharge/charge plateaus and a Coulombic efficiency close to 100% (Fig. 6). Self-discharge test of the Mg/$I_2$ battery using the MPC/$I_2$ cathode shows negligible open circuit voltage drop for 36 h (Supplementary Fig. 14), indicating the strong $I_2$ entrapment by using MPC. Further work on tailoring the physical/chemical properties of the host for better $I_2$ entrapment is still ongoing. In theory, all effective methods used in Li/S batteries can be adopted in Mg/$I_2$ batteries. For example, increasing the salt concentration in electrolytes can effectively reduce the shuttle effect and increase the Coulombic efficiency (Fig. 3a). Moreover, because of the high reversibility of

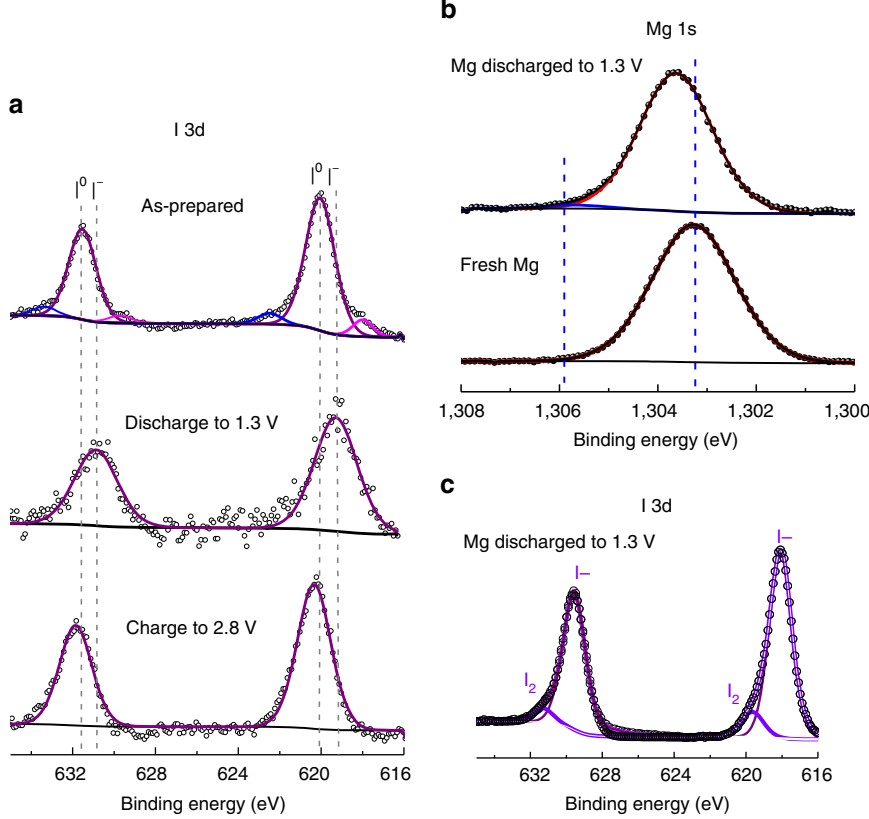

**Figure 5 | X-ray photoelectron spectroscopy study of the cathode and anode.** (**a**) High-resolution I 3d spectra of the ACC/I$_2$ cathode. (**b**) High-resolution Mg 1s spectra and (**c**) high-resolution I 3d spectrum of the Mg anode.

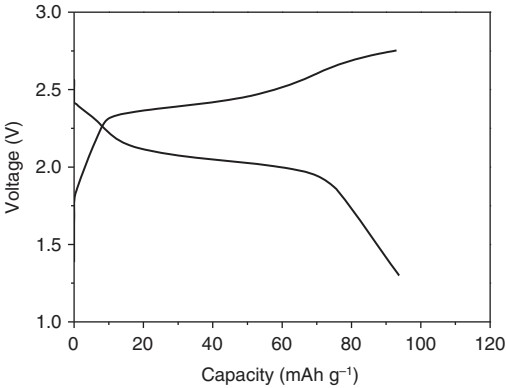

**Figure 6 | Electrochemical performance with microporous carbon/iodine cathode.** The discharge/charge curve of Mg/I$_2$ battery with MPC/I$_2$ cathode at 0.25 C.

the I$_2$ redox couple in the Mg-HDMS electrolyte and the high solubility of I$_2$ and polyiodide, a semiflow battery can be constructed based on the Mg/I$_2$ chemistry. The flooded cell results (Fig. 4a) have justified this feasibility and detailed work is also ongoing in our lab.

In summary, we demonstrate a rechargeable Mg/I$_2$ chemistry. Compared with traditional RMBs using intercalation cathodes, the I$_2$ cathode provides a high operating voltage ($\sim$2.0 V) and a much higher obtainable energy density ($\sim$400 Wh kg$^{-1}$). More importantly, the liquid–solid reaction ensures remarkable reaction kinetics and reversibility mainly because of the absence of the solid-state Mg$^{2+}$ diffusion that has been considered as a major hurdle for the development of cathode materials for RMBs. The shuttle effect because of the high solubility of I$_2$ and polyiodide

can be effectively mitigated through host optimization and/or electrolyte optimization. The chemical insights obtained in this work could guide the future design of rechargeable Mg/I$_2$ battery or semiflow battery. Above all, the demonstration of this proof-of-concept Mg/I$_2$ may open an avenue towards the development of high-performance RMB cathodes by utilizing soluble redox couples whose reactions do not rely on solid-state Mg$^{2+}$ diffusion.

## Methods

**Solubility measurement.** Sufficient I$_2$ and MgI$_2$ were added into TEGDME to form saturated solution. I$_3^-$, the most common polyiodide species, was made by adding a mixture of I$_2$/MgI$_2$ (I$_2$:MgI$_2$ = 2:1) into the solvent.

**Cathode fabrication.** ACC/I$_2$ cathode was prepared through a melt-diffusion method following a previous report[19,20]. The ACC samples (ACC-507-20) were obtained from Kynol Inc. (USA) and were cut to circular discs with a diameter of $\sim$8 mm. Elemental I$_2$ (99.98%, Sigma-Aldrich) was spread on the bottom of a stainless reactor and then ACC disks were laid on top of the I$_2$. The reactor was then sealed and heated to 135 °C for 12 h. I$_2$ loading was measured by subtracting the mass of blank ACC from the loaded ACC.

**Electrolyte preparation.** Electrolytes were prepared under pure argon atmosphere in VAC, Inc. glove box (<1 p.p.m. of water and oxygen). The non-nucleophilic Mg electrolyte based on (HMDS)$_2$Mg was synthesized following a previously reported procedure[35]. Then, 3.45 g of (HMDS)$_2$Mg was dissolved in 20 ml tetraglyme (TEGDME) with stirring for 24 h. After that, 2.67 g of AlCl$_3$ was added slowly into the solution and stirred for 24 h at room temperature. Subsequently, $\sim$0.95 g of MgCl$_2$ was added slowly to the solution and stirred for 48 h.

**Electrochemistry.** Mg foil was used as anode. Separators were Whatman Glass fiber or W-scope COD 16. Inconel alloy rod was used as current collector. For each cell, 100 μl electrolyte was added. Galvanostatic tests were carried out in Swagelok cell with Arbin Instrument. All applied current was based on the mass of active material (I$_2$). 1C rate corresponded to a current density of 211 mA g$^{-1}$$_{iodine}$. All

capacities were calculated based on the mass of active material ($I_2$) unless otherwise specified.

**Conductivity measurement.** Commercial $MgI_2$ powder was hand-milled then compressed into thin pellet. After that, a thin layer of Au was sputtered onto each side of the pellet. Then, a Swagelok cell was made by sandwiching the pellet between two ion blocking electrodes (stainless steel). Electrochemical impedance spectroscopy was conducted on the cell in the frequency range of $10^6$–1 Hz and the high-frequency intercept was read as the ionic resistance. Multiple measurements were performed and their average was calculated. Linear scanning voltammeter was performed in −0.5 to 0.5 V to extract the electronic conductivity. The potential–current curve shows a linear pattern and its slope was fitted as the electronic resistance.

**Material characterization.** X-ray diffraction patterns were obtained on Bruker Smart 1000 (Bruker AXS Inc., USA) using CuK$_\alpha$ radiation. The ACC/$I_2$ was measured with the thermogravimetric analysis equipment (SDT Q600, TA Instruments) and heated in argon atmosphere with a heating rate of $10\,°C\,min^{-1}$. The discharged sample was sealed by a plastic tape from exposure to air and moisture. SEM imaging was conducted using a Hitachi SU-70 field emission SEM. XPS analysis was measured using a Kratos Axis 165 spectrometer. Measurements were performed both before and after argon sputtering.

**Calculations.** The Gibbs formation energy of $MgI_2$ at standard conditions (298 K, 1 atm) can be calculated (data from NIST webbook)

$$\Delta G_f = \Delta H_f - T\Delta S$$
$$= -366.94\,\frac{kJ}{mol} - 298K \times (-129.67)\frac{J}{K \cdot mol} \quad (3)$$
$$= -406.6\,\frac{kJ}{mol}$$

Complete $I_2$ reduction is accompanied by 2 $e^-$ transfer per $I_2$. Therefore, the theoretical capacity of $I_2$ is

$$C = \frac{nF}{M} = \frac{2 \times 26,800\,\frac{mAh}{mol}}{2 \times 126.9\,\frac{g}{mol}} = 211.2\,\frac{mAh}{g} \quad (4)$$

The electromotive force (e.m.f.) of the $Mg/I_2$ battery is

$$V = -\frac{\Delta G_f}{nF} = \frac{406.6\,\frac{kJ}{mol}}{2 \times 96,485\,\frac{C}{mol}} = 2.1V \quad (5)$$

The theoretical energy density of $I_2$ cathode is

$$E = C \times V = 211.2\,\frac{mAh}{g} \times 2.1V = 443.5\,Wh/kg \quad (6)$$

The theoretical energy density of the $Mg/I_2$ battery based on the total electrode mass is

$$E = -\frac{\Delta G_f}{M} = \frac{406.6\,\frac{kJ}{mol}}{278.1\,\frac{g}{mol}} = 1,462.1\,\frac{kJ}{kg} = 406.1\,\frac{Wh}{kg} \quad (7)$$

**Data availability.** The data that support the findings of this study are available from the corresponding authors on request.

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

## Acknowledgements

We thank Dr Karen J. Gaskell at the Surface Analysis Center of University of Maryland for the help on the XPS test and data analysis. This work was supported as part of the Nanostructures for Electrical Energy Storage (NEES), an Energy Frontier Research Center funded by the US Department of Energy, Office of Science, Basic Energy Sciences under Award number DESC0001160. This work was also financially supported by the National Natural Science Foundation of China (Grant No. 51371186 and No. 51504234), the 'Strategic Priority Research Program' of the Chinese Project Academy of Science (Grant no. XDA09010201), Zhejiang Province Key Science and Technology Innovation Team (Grant no. 2013TD16), Zhejiang Provincial Natural Science Foundation of China

(Grant No. LY16E040001) and Ningbo 3315 International Team of Advanced Energy Storage Materials.

## Author contributions

H.T. and T.G. proposed the concept of rechargeable $Mg/I_2$ battery and did the main measurements and analysis. X.L. and X.W. prepared the electrolyte together with H.T. X.F. did the X-ray diffraction (XRD) measurements and, together with C.L., analysed the FT-IR results. C.Y. and L.S. discussed the results and contributed to preparing the manuscript. Z.M. did some part of SEM characterizations and analysis. W.H. and C.W. supervised all the experiments and calculations. All authors contributed to the discussion.

## Additional information

**Competing financial interests:** The authors declare no competing financial interests.

