## [Peer Review File · Nature Communications]

Reviewers' comments:

Reviewer #1:

Reviewer's report on manuscript NCOMMS-16-13755-T, entitled "High Power Rechargeable Mg/I₂ Battery Chemistry" by Prof Wang and colleagues.

In general this is an interesting paper that reflects highly original work on a new type of rechargeable magnesium batteries with Mg metal anodes and iodine adsorbed to activated carbon cathodes. This work may deserve publication in Nature Communication, however, there are a lot of corrections required, before a revised paper can be considered for publication.

Here are some important comments:

1. All the abbreviations should be explained in the text (even ACC, HMDS and more).
2. Figure 4S is misleading and confusing. What is the neat ACC contribution to the capacity? Is it that of ACC alone? With what active mass? What is the electrochemical reaction here? Why the charge & discharge processes differ in their capacity?
3. The electrolyte solution used has to be described in a way that its electrochemical window and Mg deposition/dissolution processes efficiency will be clear. The authors have to show that Mg anodes are fully reversible in the solutions used (cycling efficiency close to 100% should be demonstrated).
4. Convincing cycling efficiency values of full cells (have to be close to 100%) should be provided.

5. I do not believe the energy density calculations! The appropriate energy density comparison between the system described herein and Mg/solution/Mo₆S₈ cells should be provided, with a correct calculation, including solution! In Mg cells comprising Chevrel phase cathodes the amount of solution required is minimal, just to maintain electrochemical contact between the Mg metal anode and CP intercalation cathode.

6. Here, the battery includes porous carbonaceous matrix which has to contain iodine and a lot of solution in the porous structure. Upon discharge, Mg(I₃)₂ is formed in solution phase. A critical amount of solution has to be included, in order to contain the iodide salt. The system described herein does not have the advantages of “rocking chair” types batteries, in terms of requiring low solution volume.

7. There are too many mistakes in the language and descriptions, along the paper. Please see a partial list of examples below. The paper needs professional editing before being resubmitted.

Line 2: “attention” instead of “attentions”

Line 32: “...and also benefits exploring...” This sentence needs to be reworked.

Line 39: “electrolytes” instead of “electrolyte”

Line 40: “little progress” instead of “few progress”

Line 43: “cations” instead of “cation”

Line 52: “RMB’s” instead of “RMB”

Line 53: “...a Mg anode...”

Line 56: “limits” instead of “limited”

Line 57 “...and thus...”

Line 59: “up until now”

Line 63: remove “as”

Line 64: "For Mg batteries..." the whole sentence should be rewritten.

Line 73: "demonstrate" instead of "demonstrated"

Line 74: "capacity close to the theoretical value" instead of "close to theory capacity"

Line 76: "a capacity"

Line 78 "the electrolyte"

Line 94: "towards" instead of "toward"

Line 98: "the ACC" "a capillary"

Line 101: "the surface"

Line 103: "the iodine" "suggesting that iodine"

Line 105: "the electrolyte"

Line 108: the whole sentence should be rewritten

Line 109: "the ACC/12"

Line 110: "of electrolyte"

Line 114: remove "after", put a comma instead

Line 121: "...show discharge behavior similar to low concentration..."

Line 123: "In high concentrations, the ACC cathode show..."

Line 124: I guess you mean "spike" instead of "pike"?

Line 127: "retarded" might consider changing this word.

Line 128: Cyclic voltammetry is performed on a WE (in this case ACC/12), not on a cell or battery.

Rephrase the sentence.

Line 130: "accordingly" does not fit in this sentence. Change it

Line 134: "C/2" is mentioned in succession, please rephrase.

Line 137: "at a high rate of 1C"

Line 139: "densities" instead of "density"

Line 140: "RMB's"

Line 145: "solvents"

Line 146: "a solute"

Line 147: "remains" instead of "maintains"

Line 148: "a current"

Line 154: "a full" instead of "fully"

Line 155: "in" instead of "on the"

Line 158: "The electrolyte was analyzed at different states during discharge, using..."

Line 161: "In addition..." rephrase the whole sentence, simplify it.

Line 166: "to the reduction"

Line 166: "The FT-IT results..." rewrite the sentence.

Line 169: "changes"

Line 170: "the ACC" "the Mg anode"

Line 172: " the high resolution of iodine"

Line 175: see 154

Line 178: see 154

Line 181: "discharge"

Line 182: "the Mg anode"

Line 183: "discharge" "the Mg"

Line 184: "the FT-IR"

Line 185: "the following mechanism"

Line 188: remove "and"

Line 189: change "avoidance"

Line 190: "shuttle effects are inevitable and lead to..." "efficiencies"

Line 204: "effect" instead of "reaction"

Line 204: "On the other side" this phrase is inappropriate here.

Line 207: "have" instead of "has" "detailed"

Line 213: "a much higher"

Line 214: "to commercial" "chemistries"

Line 216: "independence" change it. Inappropriate

Line 216: "a major"

Line 217: "As a results of the high solubility of iodine and polyiodide, the shuttle effect can be..."

End

a. Summary of the key results:

This paper describes new prototypes of rechargeable Mg batteries, based on iodine adsorbed to activated carbon cathodes.

B.Originality and interest: if not novel, please give references.

The work is very novel and original.

C.Data & methodology: validity of approach, quality of data, quality of presentation.

See my attached report. The approach's validity needs more explanations.

D.Appropriate use of statistics and treatment of uncertainties.

See the attached report.

E.Conclusions: robustness, validity, reliability

Reasonable, but there are several questions, as reflected by my attached report.

F.Suggested improvements: experiments, data for possible revision

See my report. A lot of corrections are required.

G.References: appropriate credit to previous work?

Fine.

H.Clarity and context: lucidity of abstract/summary, appropriateness of abstract, introduction and conclusions

Reviewer #2 :

Dear Authors,

In the present manuscript, the authors proposed the novel Mg/I₂ battery system. The concept itself is interesting and the authors succeeded the concept verification. However, I think the proposed system has fatal problems as follows.

1. The proposed system shows good reversibility if the I₂ and Mg(I₃)₂ are dissolved in the electrolyte solution, however the dissolution of I₂ or Mg(I₃)₂ initiates shuttle reaction. (The authors proposes it can be prevented by using optimized MPC/I₂ composite, but no convincing data is presented. Figure 6 reminds that only the small amount of the iodine loaded in the MPC worked as the active material, and most of the capacity is due to the MPC itself.) In addition, since large concentration change of the electrolyte solution initiates the large volume change, the present system needs to have extra space for the additional electrolyte solution to avoid the dry out of the electrode. The additional electrolyte solution significantly reduces the volumetric energy density of the cell and enhances the dissolution of I₂ and Mg(I₃)₂.
2. The morphology change of the iodine in Figure 4 clearly suggests that once the iodine is dissolved, the morphology of the cathode active material is uncontrollable at this moment. The big morphology change is one of the biggest problems in the conversion-based system, I think. In the present manuscript, the authors have not pointed it out and solved the problem. I understand some electrochemical reaction like this system behaves as a good battery system, but the biggest problem behind the reaction is ignored in the manuscript. The deposition process of I₂ needs to be discussed before proposing as a battery system.
3. The XPS spectra for Mg negative electrode shown in Figure 5 (c) suggests that the I₂ dissolved in the electrolyte solution forms MgI₂ at the surface. In the case of the Mg negative electrode, fresh surface of Mg metal continuously appears during the charging-discharging process. Therefore, the I₂ in the electrolyte solution is continuously consumed. In order to reuse the I₂ at the surface of the Mg negative electrode, the potential of the negative electrode needs to be increased above 1.3 V vs. Mg. The required potential is not practical to use in the practical battery system.
4. The kinetics of the reaction is supposed to be affected by the activity of I₂ and Mg(I₃)₂ in the electrolyte solution, however discussions concerning the kinetics of the reaction is missing in the present manuscript.

5. The energy density plot in Figure 1 a) is misleading. In the present system the solvent molecules need to be considered as a part of the active materials, if the authors think the liquid-solid two phase reaction is the key of the reaction.

Based on the above comments for the authors, I would suggest to reject the present manuscript. I think a lot of the fundamental studies are necessary before proposing a novel battery system.

Point-to-Point Responses (in *italic green font*)

Reviewer #1 (Remarks to the Author):

In general this is an interesting paper that reflects highly original work on a new type of rechargeable magnesium batteries with Mg metal anodes and iodine adsorbed to activated carbon cathodes. This work may deserve publication in Nature Communication, however, there are a lot of corrections required, before a revised paper can be considered for publication.

We thank the reviewer for his/her positive comment on the originality and importance of our work.

Here are some important comments:

1. All the abbreviations should be explained in the text (even ACC, HMDS and more).
We thank the reviewer for pointing this out. We have corrected the issue and explained all abbreviations in the text the first time they are used.

2. Figure 4S is misleading and confusing. What is the neat ACC contribution to the capacity? Is it that of ACC alone? With what active mass? What is the electrochemical reaction here? Why the charge & discharge processes differ in their capacity?

We apologize for the confusion caused by the ambiguity in language and data presentation. For a typical ACC/I₂ composite cathode used in the electrochemical tests, the iodine loading is ~ 1.0 mg/cm² and carbon mass is 7.8 mg/cm². Since active carbon is known for its surface adsorption of ions during electrochemical discharge, it is necessary to exclude the contribution from this capacitive storage from the overall measured capacity. By doing so we will be able to assess the utilization of active material, i.e. iodine, during the discharge/charge process and thus analyze its reaction route.

To better interpret the results and clarify the confusion, we replot the raw discharge/charge data of blank (or neat) ACC (Figure R1). The capacity is calculated based on the mass of ACC. The sloping behavior suggests a typical electrochemical double layer storage mechanism. A capacity of ~14 mAh/g-carbon can be obtained when discharged to 1.3V and ~8 mAh/g-carbon when discharged to 1.6V. If this capacitive storage is misinterpreted as the capacity from iodine redox reaction, based on the mass ratio of carbon/iodine, it corresponds to $\frac{8 \text{ mAh}}{\text{g}} \times \frac{7.8 \frac{\text{mg}}{\text{cm}^2}}{1 \frac{\text{mg}}{\text{cm}^2}} = 62 \text{ mAh/g}$ for samples discharged to 1.6V (Figure 3a) and 109 mAh/g for samples discharged to 1.3V (Figure S5). This simple analysis demonstrates that the contribution of carbon to the overall capacity (Figure 3a, 241 mAh/g; Figure S5, 308 mAh/g) cannot be neglected. When calculating the capacity and utilization of I₂, we intentionally subtract the contribution of carbon capacitive storage from the measured capacity.

Figure R1. Discharge/charge curve of blank (or neat) ACC in 0.5 M Mg-HMDS electrolyte at a current density of 52.7 $\mu\text{A}/\text{cm}^2$ and in the voltage window of 1.3-2.9V

Since the capacitive storage is usually very reversible, the discharge capacity is almost the same with the charge capacity.

In the supporting information, we have updated Figure S4 with related discussion.

3. The electrolyte solution used has to be described in a way that its electrochemical window and Mg deposition/dissolution processes efficiency will be clear. The authors have to show that Mg anodes are fully reversible in the solutions used (cycling efficiency close to 100% should be demonstrated).

We thank the reviewer for the valuable comments. The basic properties of the electrolyte have been well documented in literature by Karger et al.^[1] Therefore, we did not add the data of electrochemical window and Mg deposition/dissolution in our previous manuscript. We apologize for not making this clear in the manuscript.

Here we provide our results for clarifying any questions regarding the electrolyte. First, cyclic voltammeter (CV) test was conducted in a three-electrode set-up to illustrate the Mg deposition/dissolution process and its electrochemical stability window (Figure R2). During the cathodic scan, Mg deposition starts at -0.5V and deposition current increases rapidly with increasing overpotential. During the reverse scan, Mg stripping starts at -0.16 V. The Coulombic Efficiency of the deposition/stripping process can be evaluated by integrating current with time (Figure R2b). The Coulombic efficiency of 2M Mg-HMDS in TEGDME electrolyte is a value of 94.5%. The electrolyte is quite stable

until the voltage reaches $> 2.7V$ when oxidative current can be observed. Strong decomposition occurs when voltage exceeds $3.0V$.

Figure R2. a) CV of electrolyte(2M Mg-HMDS in TEGDME). Scan rate: 100 mV/s, voltage:-1V-3.5V. Working electrode: Pt, Reference electrode: Mg foil, Counter electrode: Mg foil. b) capacity-time curve

We further measured the Mg deposition/stripping process in coin cells using Cu foil as working electrode and Mg foil as anode(Figure R3). Figure R3a gives the galvanostatic discharge/charge curve. In each cycle Mg was deposited onto Cu foil at a current density of 0.1 mA/cm^2 for 1h, and then stripping occurs at the same current until voltage reaches 1V. A typical discharge(deposition) and charge(stripping) curve is given in Figure R3b. During deposition, after the initial large overpotential to initiate the nucleation process, voltage stabilize at an overpotential below 0.1 V. During stripping, overpotential is also below 0.1 V until the end of the charge. Figure R3c gives the Coulombic efficiency during cycling. After the initial activation cycles, an efficiency of close to 100% can be achieved.

In summary, consistent with previous report,^[1] the Coulombic efficiency of the electrolyte can achieve >95% with different testing methods, which is close to 100%. In terms of electrochemical stability window, the electrolyte may start to decompose above 2.7V. However, strong decomposition will not be seen until 3.0V.

We have added Figure R2 and R3 into the revised supporting information as Figure S3 and S4, respectively.

Figure R3. a) Galvanostatic discharge/charge of Cu|Mg coin cell. Current: 0.1 mA/cm². Discharge time: 1h. Charge cut-off voltage: 1V; b) A typical discharge/charge curve; c) Coulombic Efficiency for Mg depositing/stripping on Cu foil

4. Convincing cycling efficiency values of full cells (have to be close to 100%) should be provided.

We thank the reviewer for his suggestion. The Coulombic Efficiency of the full cell is given in Figure R4. It shows very high Coulombic Efficiency. In the revised manuscript, the result was added into Figure 3c.

Figure R4. Coulombic Efficiency and Cycling stability of Mg/I₂ full cell

5. I do not believe the energy density calculations! The appropriate energy density comparison between the system described herein and Mg/solution/Mo₆S₈ cells should be provided, with a correct calculation, including solution! In Mg cells comprising

Chevrel phase cathodes the amount of solution required is minimal, just to maintain electrochemical contact between the Mg metal anode and CP intercalation cathode.

We thank the reviewer for the comment. We agree that for a rocking-chair system using intercalation compounds (such as Mo_6S_8), the mass of electrolyte accounts for a small portion of the total battery mass since electrolyte only functions as ion conductor. A multi-phase battery system, in which intermediate products are dissolved into electrolyte, has different requirement regarding the amount of electrolyte.

We want to answer this puzzle by first emphasizing that our system is intrinsically similar to a Li/S system in reaction mechanism. Both systems form soluble intermediate products and insoluble solids as final product. In Li/S system, the discharge process is accompanied by two consecutive equilibria of sulfur species: $\text{S}_8\text{-Li}_2\text{S}_{n, n=8-4}$ (Solid-Liquid) and $\text{Li}_2\text{S}_{n, n=8-4}\text{-Li}_2\text{S}_{n, n=2-1}$ (Liquid-Solid). In each equilibrium, both soluble and insoluble sulfur species co-exist.^[2] Fast kinetics can be obtained on condition that some sulfur species dissolves, so that the two-phase reaction is maintained. In other words, not all sulfur species exist in liquid phase during the discharge/charge of a Li/S battery as a result of the dynamic equilibria of multiple sulfur species.^[3] Therefore, the amount of electrolyte can be optimized to a minimum value as long as the free diffusion of dissolved polysulfide species is enabled.^[5] In fact, extensive studies have shown that in concentrated electrolyte, in which the solubility of polysulfide is low but not zero, Li/S batteries can also work with fast kinetics.^[4] This further confirms that fast kinetics only require a small amount of sulfur species to be dissolved. For the similar reason, electrolyte/iodine ratio can also be optimized in the Mg/I_2 battery for maintaining its fast kinetics while lowering the amount of electrolyte.

Even though sufficient electrolyte is required to dissolve all iodine species, the Mg/I_2 system can still be competitive to the current rocking-chair magnesium battery, as we have demonstrated in the following analysis. For rocking-chair systems, Hagen et al. has disassembled a commercial 18650 battery comprising of LiCoO_2 cathode and graphite anode and systematically measured the mass of different cell components.^[6] He found that electrode material takes 61.1% (cathode 36.9% and anode 24.2%) of the battery weight, while electrolyte only takes 9.1%. This means that electrolyte mass is only 14.9 % of cathode and anode material combined. If considering the total weight of cathode material, anode material and electrolyte, and neglecting packaging, current collector, separator and other inactive cell components, the energy density of a rocking chair system can be calculated by dividing the energy density of the redox couple by a factor of 1.15 (i.e. $1+9.1\%/61.1\%$). Consequently, the energy density of $\text{Mo}_6\text{S}_8/\text{Mg}$ system is $126/1.15=109.5$ Wh/kg.

For the Mg/I_2 system, the solubility of iodine in TEGDME is > 75 g/100 mL. For 1 mol iodine(254 g), 338 mL or 341 g of TEDGME is required, which add an extra 123% weight to the system other than the mass of electrode material(cathode: 254 g of iodine, anode 24 g of Mg). Thus the energy density of a Mg/I_2 battery can be calculated by dividing the energy density of the redox couple by a factor of 2.23($1+ 123\%$). Consequently, the energy density of the Mg/I_2 system is $367/2.23=164.7$ Wh/kg. As can

be seen, it is still higher than the energy density of a $\text{Mo}_6\text{S}_8/\text{Mg}$ battery (109.5 Wh/kg). Note, the solubility of iodine in TEGDME is > 75 g, indicating that advantage of the Mg/I_2 system over a $\text{Mo}_6\text{S}_8/\text{Mg}$ system can be larger than the calculation here.

In summary, similar to a Li/S system, the amount of electrolyte for the Mg/I_2 system can be optimized to a minimum value while maintaining the fast kinetics. Furthermore, the Mg/I_2 system can be competitive to current rocking-chair magnesium battery even we consider the mass of electrolyte.

6. Here, the battery includes porous carbonaceous matrix which has to contain iodine and a lot of solution in the porous structure. Upon discharge, $\text{Mg}(\text{I}_3)_2$ is formed in solution phase. A critical amount of solution has to be included, in order to contain the iodide salt. The system described herein does not have the advantages of “rocking chair” types batteries, in terms of requiring low solution volume.

We thank the reviewer for the comments. It is not necessary to dissolve all the intermediate iodine species (polyiodide) for achieving the fast kinetics in the Mg/I_2 cells. In fact, we want to dissolve only a small amount of polyiodide to achieve both the fast reaction kinetics and slow shuttle reaction, which is exactly the same as in Li/S batteries. Regarding how the mass of electrolyte would affect the energy density of our Mg/I_2 battery, please see the answer to question 5.

7. There are too many mistakes in the language and descriptions, along the paper. Please see a partial list of examples below. The paper needs professional editing before being resubmitted.

We are grateful for the valuable comments from the reviewer, and we apologize for the problems in the language. All the mentioned problems have been corrected according to the suggestion. The resubmitted paper was carefully reviewed by native speaker with related expertise.

Reviewer #2 (Remarks to the Author):

In the present manuscript, the authors proposed the novel Mg/I_2 battery system. The concept itself is interesting and the authors succeeded the concept verification. However, I think the proposed system has fatal problems as follows.

We thank the reviewer for the comment, and appreciate his understanding of the originality of our work and the concept of this novel battery system. However, we disagree with the reviewer on the fatal problems as discussed below.

1. The proposed system shows good reversibility if the I_2 and $\text{Mg}(\text{I}_3)_2$ are dissolved in the electrolyte solution, however the dissolution of I_2 or $\text{Mg}(\text{I}_3)_2$ initiates shuttle reaction.

(The authors proposes it can be prevented by using optimized MPC/I₂ composite, but no convincing data is presented. Figure 6 reminds that only the small amount of the iodine loaded in the MPC worked as the active material, and most of the capacity is due to the MPC itself.) In addition, since large concentration change of the electrolyte solution initiates the large volume change, the present system needs to have extra space for the additional electrolyte solution to avoid the dry out of the electrode. The additional electrolyte solution significantly reduces the volumetric energy density of the cell and enhances the dissolution of I₂ and Mg(I₃)₂.

We thank the reviewer for his insights regarding the shuttle reaction in our system. Similar to ether electrolyte based Li/S chemistry, dissolution of redox couple leads to fast reaction kinetics, but also causes self-discharge due to the shuttle of soluble redox species. Although the shuttle effect is an intrinsic problem for any battery chemistry that has soluble reaction intermediates, various strategies have shown promising results in mitigating the shuttle effect to the extent where practical applications is possible. These strategies include novel electrolyte compositions and additives^{[4][7][8]} and novel host structure^{[9][10]}. Microporous carbon (MPC) is one sulfur host material that is able to well prevent the dissolution of polysulfide, thus leading to remarkable cycling stability of >4000 cycles.^[11] We expect MPC can play similar role in our Mg/Iodine system. Indeed, confining iodine species within the micropores of MPC dramatically retards the shuttle effect, as indicated by the high Coulombic Efficiency (99%) for the discharge/charge curves of MPC/I₂ cathode (Figure 6). The very stable open-circuit voltage for > 36 hours for the MPC/I₂ cathode measured during storage also confirms the lack of shuttle effect.

In this work, the MPC/I₂ cathode has a typical iodine loading of 1 mg/cm², while MPC weighs 3 mg/cm². Just like other carbon material, MPC itself may provide some capacity by surface adsorption of ions. We intentionally conducted galvanostatic experiment with blank MPC cathode. The discharge/charge curve was given in Figure R5. A typical electrochemical double layer capacitor behavior can be identified with a capacity of 4 mAh/g-carbon, which is equivalent to 12 mAh/g-iodine if this capacitive storage is misinterpreted as part of iodine's redox capacity. Nevertheless, in the discharge curve MPC/I₂, a total capacity of ~95 mAh/g-iodine can be obtained, suggesting that the contribution of carbon only takes a small portion. Indeed, a discharge plateau at ~2.1 V can be clearly identified, which is a strong sign of iodine reduction instead of carbon surface adsorption.

It is highly unlikely that strong electrolyte volume change will occur during the reaction since in our solubility measurement we observed that the volume of saturated Iodine solution is only ~10% more than the volume of solvent.

Figure R5. Discharge/charge curve of blank (or neat) MPC in 0.5 M Mg-HMDS electrolyte at a current density of 52.7 $\mu\text{A}/\text{cm}^2$ and in the voltage window of 1.3-2.8V

2. The morphology change of the iodine in Figure 4 clearly suggests that once the iodine is dissolved, the morphology of the cathode active material is uncontrollable at this moment. The big morphology change is one of the biggest problems in the conversion-based system, I think. In the present manuscript, the authors have not pointed it out and solved the problem. I understand some electrochemical reaction like this system behaves as a good battery system, but the biggest problem behind the reaction is ignored in the manuscript. The deposition process of I₂ needs to be discussed before proposing as a battery system.

We thank the reviewer for the comments. The morphology change is indeed a big problem for conversion cathode materials, e.g. FeF₃ or MnO₂, because local morphology change can destroy the structure integrity of the electrode and cut off part of active material from the electronic conduction pathway, leading to capacity fade. However, for our Mg/I₂ system, 1) the morphology change the reviewer has observed is actually from carbon host instead of iodine, given the fact the iodine and polyiodide is in the solution phase and the final discharged product, the insoluble MgI₂, is distributed on the surface of carbon. 2) This morphology change of carbon is a result of long term immersing and hanging in solution instead of iodine reaction due to the weak connection between individual carbon fibers (Figure 2b). It is common for this kind of woven carbon cloth to lose their mechanical integrity when it is being soaked into a wettable liquid for long time because surface tension between individual fiber and the solution and gravity can be larger than the weak mechanical force between fibers. Actually, for all carbon/iodine cathodes tested in Swagelok cells, where the electrode is backed by a rigid current collector instead of being hanged in the electrolyte, no significant morphology change is observed (Figure R6).

Figure R6. The images of ACC/iodine cathode. left: after cycle; right: pristine. We can see through the pristine one since there are voids between the woven fibers (Figure 2b). However, in the cycled one the voids are occupied by residual electrolyte.

3. The XPS spectra for Mg negative electrode shown in Figure 5 (c) suggests that the I₂ dissolved in the electrolyte solution forms MgI₂ at the surface. In the case of the Mg negative electrode, fresh surface of Mg metal continuously appears during the charging-discharging process. Therefore, the I₂ in the electrolyte solution is continuously consumed. In order to reuse the I₂ at the surface of the Mg negative electrode, the potential of the negative electrode needs to be increased above 1.3 V vs. Mg. The required potential is not practical to use in the practical battery system.

We thank the reviewer for the comments. Similar to Li/S batteries, the dissolved polyiodide will react with Mg anode and MgI₂ will be formed on Mg surface (Figure 5c). Since magnesium iodide is an insoluble solid in the electrolyte, it will cover the surface of Mg metal and can prevent any further reaction of Mg anode. For this reason, I₂ in the electrolyte solution will not be continuously consumed. As expected when we were conceiving the Mg/I₂ battery concept, and later confirmed by the XPS results (Figure 5c), the shuttle of the soluble iodine and polyiodide will inevitably cause the loss of active material. However, the close to 100% discharge/charge Coulombic Efficiency (Figure 3a, Figure 6) and the stable cycling of the Mg/I₂ battery (Figure 3c) have well illustrated that there is no continues loss of active material during charge/discharge cycles.

Similar to Li/S battery, we admit the shuttle effect is going to be a major issue for turning this chemistry into practical products. Nevertheless, in this work our major aim is to demonstrate the feasibility of utilizing the Mg/I₂ chemistry to realize a high power rechargeable magnesium battery. The discharge/charge curves (Figure 3a, Figure 6) and cycling stability of >100 cycles (Figure 3c) have well proved the concept. We devised improved electrolyte composition and cathode host material to cope with the shuttle effect, which shows promising results. We do believe that this proof-of-concept paper can trigger the interest of magnesium battery community, thus bringing more manpower and intelligence resources into this field. The shuttle effect could be well resolved then just like scientists have done on Li/S battery in the past years.

4. The kinetics of the reaction is supposed to be affected by the activity of I₂ and Mg(I₃)₂ in the electrolyte solution, however discussions concerning the kinetics of the reaction is missing in the present manuscript.

We thank the reviewer for the comments. To study how the kinetics of the system evolve with the concentration change of iodine and polyiodide, we monitored the impedances of the system during discharge at different voltage. The battery is discharged to a certain voltage by CCCV (constant current then constant voltage) method before impedance measurement. The impedances are plotted in Figure R7. The just assembled cell shows a large charge transfer resistance of > 1000 ohm, probably due to the activation energy required for initiating iodine → polyiodide phase transition (Figure R7, OCV). However, after the initiation, charge transfer resistance drops down by one order of magnitude to several hundreds. Interestingly, the charge transfer resistance does not change significantly at different voltages. According to Bard et al,^[12] this could be a sign that the heterogeneous reaction constant (exchange current) is so large that the influence of reactants and products concentration change on kinetics is not significant. To get an accurate estimate of the charge transfer resistance, we fitted data at 1.7V and the charge transfer resistance for the iodine redox reaction is 272 ohm.

Figure R7. Electrochemical Impedance Spectrum (EIS) results of the Mg/I₂ battery during discharge at different voltage.

Meanwhile, we also conduct impedance tests with intercalation cathode Mo₆S₈ as cathode (Figure R8), to compare the kinetics between a rocking-chair Mg battery and the Mg/I₂ system. The high and medium frequency response (10⁶-0.1 Hz) was fitted into two separate semi-circles to represent the charge transfer resistance of the cathode redox reaction and anode redox reaction. An Mg/Mg symmetrical cell was built to identify the characterizing frequency for Mg deposition/stripping in this electrolyte (Figure

R9), and it was found that the characterizing frequency for Mg anode redox reaction is 0.5 Hz. According to this, the medium frequency semi-circle(0.5 Hz) in Figure R8 is assigned to Mg anode reaction in the $\text{Mo}_6\text{S}_8/\text{Mg}$ battery, and the high frequency semi-circle(100 Hz) is assigned to cathode redox reaction. From the fitting result, the cathode charge transfer resistance for the rocking chair system is 2002 ohm, which is about one order magnitude higher than that of the Mg/I_2 battery. This further confirms the fast reaction kinetics of our Mg/I_2 battery.

Figure R8. Electrochemical Impedance Spectrum (EIS) results of $\text{Mo}_6\text{S}_8/\text{Mg}$ battery

Figure R9. Electrochemical Impedance Spectrum (EIS)

results of Mg/Mg symmetrical cell

These kinetics studies were added in the support information of the revised version.

5. The energy density plot in Figure 1 a) is misleading. In the present system the solvent molecules need to be considered as a part of the active materials, if the authors think the liquid-solid two phase reaction is the key of the reaction.

We thank the reviewer for the comment. We agree that for a rocking-chair system using intercalation compounds (such as Mo_6S_8), the mass of electrolyte accounts for a small portion of the total battery mass since electrolyte only functions as ion conductor. A multi-phase battery system, in which intermediate products are dissolved into electrolyte, has different requirement regarding the amount of electrolyte.

We want to answer this puzzle by first emphasizing that our system is intrinsically similar to a Li/S system in reaction mechanism. Both systems form soluble intermediate products and insoluble solids as final product. In Li/S system, the discharge process is accompanied by two consecutive equilibria of sulfur species: $\text{S}_8\text{-Li}_2\text{S}_{n, n=8-4}$ (Solid-Liquid) and $\text{Li}_2\text{S}_{n, n=8-4}\text{-Li}_2\text{S}_{n, n=2-1}$ (Liquid-Solid). In each equilibrium, both soluble and insoluble sulfur species co-exist.^[2] Fast kinetics can be obtained on condition that some sulfur species dissolves, so that the two-phase reaction is maintained. In other words, not all sulfur species exist in liquid phase during the discharge/charge of a Li/S battery as a result of the dynamic equilibria of multiple sulfur species.^[3] Therefore, the amount of electrolyte can be optimized to a minimum value as long as the free diffusion of dissolved polysulfide species is enabled.^[5] In fact, extensive studies have shown that in concentrated electrolyte, in which the solubility of polysulfide is low but not zero, Li/S batteries can also work with fast kinetics.^[4] This further confirms that fast kinetics only require a small amount of sulfur species to be dissolved. For the similar reason, electrolyte/iodine ratio can also be optimized in the Mg/I_2 battery for maintaining its fast kinetics while lowering the amount of electrolyte.

Even though sufficient electrolyte is required to dissolve all iodine species, the Mg/I_2 system can still be competitive to the current rocking-chair magnesium battery, as we have demonstrated in the following analysis. For rocking-chair systems, Hagen et al. has disassembled a commercial 18650 battery comprising of LiCoO_2 cathode and graphite anode and systematically measured the mass of different cell components.^[6] He found that electrode material takes 61.1% (cathode 36.9% and anode 24.2%) of the battery weight, while electrolyte only takes 9.1%. This means that electrolyte mass is only 14.9 % of cathode and anode material combined. If considering the total weight of cathode material, anode material and electrolyte, and neglecting packaging, current collector, separator and other inactive cell components, the energy density of a rocking chair system can be calculated by dividing the energy density of the redox couple by a factor of 1.15 (i.e. $1+9.1\%/61.1\%$). Consequently, the energy density of $\text{Mo}_6\text{S}_8/\text{Mg}$ system is $126/1.15=109.5$ Wh/kg.

For the Mg/I₂ system, the solubility of iodine in TEGDME is > 75 g/100 mL. For 1 mol iodine(254 g), 338 mL or 341 g of TEDGME is required, which add an extra 123% weight to the system other than the mass of electrode material(cathode: 254 g of iodine, anode 24 g of Mg). Thus the energy density of a Mg/I₂ battery can be calculated by dividing the energy density of the redox couple by a factor of 2.23(1+ 123%). Consequently, the energy density of the Mg/I₂ system is 367/2.23=164.7 Wh/kg. As can be seen, it is still higher than the energy density of a Mo₆S₈/Mg battery (109.5 Wh/kg). Note, the solubility of iodine in TEGDME is > 75 g, indicating that advantage of the Mg/I₂ system over a Mo₆S₈/Mg system can be larger than the calculation here.

In summary, similar to a Li/S system, the amount of electrolyte for the Mg/I₂ system can be optimized to a minimum value while maintaining the fast kinetics. Furthermore, the Mg/I₂ system can be competitive to current rocking-chair magnesium battery even we consider the mass of electrolyte.

- [1] Z. Zhao-Karger, X. Zhao, O. Fuhr, M. Fichtner, *RSC Adv.* **2013**, 3, 16330.
- [2] S. S. Zhang, *J. Power Sources* **2013**, 231, 153–162.
- [3] Y. V. Mikhaylik, J. R. Akridge, *J. Electrochem. Soc.* **2004**, 151, A1969.
- [4] L. Suo, Y.-S. Hu, H. Li, M. Armand, L. Chen, *Nat. Commun.* **2013**, 4, 1481.
- [5] S. S. Zhang, *Energies* **2012**, 5, 5190–5197.
- [6] M. Hagen, D. Hanselmann, K. Ahlbrecht, R. Maça, D. Gerber, J. Tübke, *Adv. Energy Mater.* **2015**, n/a–n/a.
- [7] a. Rosenman, R. Elazari, G. Salitra, E. Markevich, D. Aurbach, a. Garsuch, *J. Electrochem. Soc.* **2015**, 162, A470–A473.
- [8] E. Markevich, G. Salitra, a. Rosenman, Y. Talyosef, F. Chesneau, D. Aurbach, *J. Mater. Chem. A* **2015**, 3, 19873–19883.
- [9] X. Ji, S. Evers, R. Black, L. F. Nazar, *Nat. Commun.* **2011**, 2, 325.
- [10] X. Liang, C. Hart, Q. Pang, A. Garsuch, T. Weiss, L. F. Nazar, *Nat. Commun.* **2015**, 6, 5682.
- [11] Y. Xu, Y. Wen, Y. Zhu, K. Gaskell, K. A. Cychosz, B. Eichhorn, K. Xu, C. Wang, *Adv. Funct. Mater.* **2015**, 25, 4312–4320.
- [12] A. Bard, L. Faulkner, *Electrochemical Methods: Fundamentals and Applications*, **2000**.

Reviewers' comments:

Reviewer #1 chose only to provide confidential comments to the editor.

Reviewer #2 (Remarks to the Author):

Thank you very much for the reply and the detailed explanations. Now I well understand the advantage and disadvantages of the Mg/I₂ system as the authors pointed out.

However I still think the system has a couple of fundamental problems. I think the manuscript might be deserved to be published in Nature Communications if the following points are clearly explained.

1. I₂ entrapment capability of the carbonaceous materials is very crucial in the present system. The figure S1 and the figure S14 show good indication of the I₂ (or poly iodide) entrapment. However I think the materials characterization of the I₂ entrapped in the ACC or MPC is missing in the manuscript. Additional quantitative analysis (iodine / carbon ratio) should also be provided to estimate the practical energy density.

2. The explanation for my comment 3 concerning the continuous I₂ loss is still not convincing. It makes sense to me that the stable MgI₂ prevents the further reduction of the I₂ or poly-iodine species. However once the Mg dissolution (= discharging) starts, the interphase between Mg and MgI₂ should be broken and fresh Mg surface appears. Then the MgI₂ formation should take place as the fresh Mg metal surface appears. Since MgI₂ is ionic crystal, the MgI₂ layer should not be able to follow the continuous morphology change of the Mg negative electrode.

I understand the authors demonstrated good cycling performance of the Mg/I₂ swagelok cell as shown in figure 3 c), but the authors do not provide enough evidence why the Mg/I₂ cell provides the good cycle ability. Some other reaction should contribute to prevent the continuous I₂ consumption. The authors should prove why the Mg/I₂ system shows the good reversibility. What is the reaction preventing the continuous I₂ consumption?

I think it is interesting work, but a couple of critical points should be proven before publication.

Reviewer #2 (Remarks to the Author):

Thank you very much for the reply and the detailed explanations. Now I well understand the advantage and disadvantages of the Mg/I₂ system as the authors pointed out. However I still think the system has a couple of fundamental problems. I think the manuscript might be deserved to be published in NatureCommunications if the following points are clearly explained.

We thank the reviewer for his positive comment and appreciate his understanding of the advantage of such a new system.

1. I₂ entrapment capability of the carbonaceous materials is very crucial in the present system. The figure S1 and the figure S14 show good indication of the I₂ (or poly iodide) entrapment. However I think the materials characterization of the I₂ entrapped in the ACC or MPC is missing in the manuscript. Additional quantitative analysis (iodine / carbon ratio) should also be provided to estimate the practical energy density.

We thank the reviewer for his comment. The morphology, element distribution, XRD pattern of the ACC/I₂ cathode were provided in Figure 2 and the weight ratio of iodine for the ACC/I₂ composite cathode was evaluated by Thermal Gravimetric Analysis in Figure S2. A typical ACC/I₂ composite cathode used in the electrochemical tests has iodine loading of ~1.0 mg/cm² and ACC loading of ~7.8 mg/cm². The ACC functions as both the I₂ host and the current collector. Since the specific capacity of ACC/I₂ is 240 mAh/g-I₂, and the average discharge voltage is 1.9V, the practical energy density of the cathode composite is $1.9 \times 240 \times 1/8.8 = 51.8$ Wh/kg (for total mass of the cathode). For a rocking-chair battery comprising of Mo₃S₄ cathode, since binder, carbon and current collector are necessary, the practical energy density is much lower than what the material can offer. Taking the work by Aurbach et al as an example^{[1][2]}, for a typical Mo₃S₄ electrode of 5 mg/cm² active material loading, 10 wt% binder and 10 wt% carbon are added. Meanwhile, stainless steel current collector has to be used to resist the corrosive electrolyte, which has a weight of ~ 10 mg/cm² (thickness 25 μm). As a result, the energy density of the cathode composite is $1.1V \times 110$ Wh/kg $\times 80\% \times 5/15 = 32$ Wh/kg. Therefore, the practical energy density of our current Mg/I₂ battery is much higher than the reported rocking chair battery.

We need to emphasize that in this paper, we focus on the demonstration of the Mg/I₂ battery concept, instead of optimizing the ACC/I₂ cathode to achieve high energy density. Nevertheless, we can still estimate the achievable energy density of ACC/I₂ with the experience from the mature ACC/S cathode due to the similarity between them. Currently, a high sulfur loading of 50% can be realized with the ACC host in Li/S battery.^[3] With such a high loading, the practical energy density of an ACC/I₂ cathode can reach $1.9 \times 240 \times 0.5 = 228$ Wh/kg, which is much higher than what the Chevrel Phase can offer in theory!

2. The explanation for my comment 3 concerning the continuous I₂ loss is still not convincing. It makes sense to me that the stable MgI₂ prevents the further reduction of the I₂ or poly-iodine species. However once the Mg dissolution (= discharging) starts, the interphase between Mg and MgI₂ should be broken and fresh Mg surface appears. Then the MgI₂ formation should take place as the fresh Mg metal surface appears. Since MgI₂ is ionic crystal, the MgI₂ layer should not be able to follow the continuous morphology change of the Mg negative electrode.

I understand the authors demonstrated good cycling performance of the Mg/I₂ swagelok cell as shown in figure 3 c), but the authors do not provide enough evidence why the Mg/I₂ cell provides the good cycle ability. Some other reaction should contribute to prevent the continuous I₂ consumption. The authors should prove why the Mg/I₂ system shows the good reversibility. What is the reaction preventing the continuous I₂ consumption?

We thank the reviewer for his insightful comments. In Li/S system, the formed Li₂S-based passivation layer on Li anode functions as a solid electrolyte interphase (SEI) which conducts Li⁺ but blocks the direct contact of polysulfide with Li anode.^{[4][5]} It is highly likely that the formed MgI₂ functions in a similar way to conduct Mg ion and also prevent vigorous reactions between Mg anode and polyiodide in our Mg/I₂ battery for the following reasons.

First, the presence of MgI₂ on Mg anode in the cycled cells have been proved by high resolution XPS spectrum of Mg and I. (Figure 5b-c)

Second, we measured the ionic and electronic conductivity of MgI₂. Commercial MgI₂ powder was hand-milled then compressed into thin pellet. After that, a thin layer of Au was sputtered onto each side of the pellet. Then a Swagelok cell was made by sandwiching the pellet between two ion blocking electrodes (stainless steel). Electrochemical impedance spectroscopy (EIS) was conducted on the cell in the frequency range of 10⁶-1 Hz and the high frequency intercept was read as the ionic resistance. Multiple measurements were performed and an average ionic conductivity of 2.0×10⁻⁵ S/cm was obtained. The value is consistent with previous study.^[6] Linear scanning voltammeter was performed in -0.5V-0.5V to extract the electronic conductivity. The potential-current curve shows a linear pattern and its slope was fitted as the electronic resistance. An electronic conductivity of 2.1×10⁻⁹ S/cm can be obtained. These results have confirmed that MgI₂ can function as a solid electrolyte, in which ionic conductivity is much higher (in this case, 4 orders higher) than electronic conductivity. The electronic and ionic conductivities and related discussion were added in the revised manuscript.

Based on the analysis above, we have reasons to believe the leaking of iodine from cathode will lead to the formation of MgI₂ solid layer, which can function as SEI that can prevent further reaction of polyiodide with Mg. During charge/discharge, Mg ion can migrate through the surface layer and participate in the charge transfer reaction, ensuring Mg deposition/stripping.

As for the mechanical property of the surface layer, the MgI₂ layer should be able to accommodate morphology change at low cycling rate, similar to LiF, which is the main constituent of SEI formed in lithium ion battery^{[7][8]} and also an ionic compound. Even through the MgI₂ layer breaks due to the volume change of Mg anode during Mg dissolution/deposition, the fresh Mg will react I₂ or polyiodine species to form MgI₂, self-healing the cracks. This self-healing mechanism of SEI has been well known in Li batteries and Na batteries. Since Mg prefers 2D uniform deposition and does not form dendrite^{[9][10]}, MgI₂ should be much more stable than the surface layers on Li and Na anodes.

- [1] D. Aurbach, Z. Lu, a Schechter, Y. Gofer, H. Gizbar, R. Turgeman, Y. Cohen, M. Moshkovich, E. Levi, *Nature* **2000**, 407, 724–727.
- [2] D. Aurbach, G. S. Suresh, E. Levi, A. Mitelman, O. Mizrahi, O. Chusid, M. Brunelli, *Adv. Mater.* **2007**, 19, 4260–4267.
- [3] R. Elazari, G. Salitra, A. Garsuch, A. Panchenko, D. Aurbach, *Adv. Mater.* **2011**, 23, 5641–5644.
- [4] S. S. Zhang, *Electrochim. Acta* **2012**, 70, 344–348.
- [5] D. Aurbach, E. Pollak, R. Elazari, G. Salitra, C. S. Kelley, J. Affinito, *J. Electrochem. Soc.* **2009**, 156, A694.
- [6] A. Hanom Ahmad, F. S. Abdul Ghani, *AIP Conf. Proc.* **2009**, 1136, 31–35.
- [7] K. Xu, *Chem. Rev.* **2014**, 114, 11503–11618.
- [8] Y. Yamada, A. Yamada, *J. Electrochem. Soc.* **2015**, 162, A2406–A2423.
- [9] M. Matsui, *J. Power Sources* **2011**, 196, 7048–7055.
- [10] C. Ling, D. Banerjee, M. Matsui, *Electrochim. Acta* **2012**, 76, 270–274.